# Prevalence of polypharmacy and the association with non-communicable diseases in Qatari elderly patients attending primary healthcare centers: A cross-sectional study

**Ayman Al-Dahshan**[1]*, **Noora Al-Kubiasi**[2], **Manal Al-Zaidan**[3], **Wael Saeed**[3], **Vahe Kehyayan**[4], **Iheb Bougmiza**[2]

1 Community Medicine Residency Program, Hamad Medical Corporation, Doha, Qatar, 2 Community Medicine Residency Program, Primary Health Care Corporation, Doha, Qatar, 3 Clinical Affairs Department, Primary Health Care Corporation, Doha, Qatar, 4 University of Calgary in Qatar, Doha, Qatar

* Ayman.aldahshan@hotmail.com

**Data Availability Statement:** All relevant data are within the manuscript and its Supporting Information files.

## Abstract

### Background

Polypharmacy has become a global public health concern particularly in the elderly population. The elderly population is the most susceptible to the negative effects of polypharmacy due to their altered pharmacokinetics and decreased drug clearance. Therefore, polypharmacy can lead to poor health status and higher rates of morbidity and mortality.

### Objective

The objective of this study was to determine the prevalence of polypharmacy ($\geq$ 5 drugs) and its association with non-communicable diseases (NCDs) in elderly ($\geq$65 years) Qatari patients attending Primary Healthcare (PHC) centers in Qatar.

### Methods

A retrospective cross-sectional analysis was conducted using the Electronic Medical Record (EMR) database of all PHC centers in Qatar for six months (April-September 2017).

### Results

Out of 5639 patients screened, 75.5% (95% CI: 74.3–76.6) were exposed to polypharmacy. Females were 1.18 times more likely to have polypharmacy compared to males (95% CI: 1.03–1.34). The multivariate analysis identified having hypertension (AOR 1.71; 95% CI: 1.38–2.13), diabetes (AOR 2.38; 95% CI: 1.97–2.87), dyslipidemia (AOR 1.29; 95% CI: 1.06–1.56), cardiovascular disease (AOR 1.56; 95% CI: 1.25–1.95) and asthma (AOR 1.39; 95% CI: 1.13–1.72) to be independent parameters associated with polypharmacy. Also, the Body Mass Index (BMI) and number of NCDs were found to be significant independent parameters associated with polypharmacy.

**Funding:** The author(s) received no specific funding for this work.

**Competing interests:** The authors have declared that no competing interests exist.

## Conclusions

The prevalence of polypharmacy among Qatari elderly attending PHC Centers is very high. Our findings confirm the strong relationship between polypharmacy and BMI, and certain NCDs. Healthcare professionals should be educated about the magnitude of polypharmacy, its negative effects, and its associated factors. Best practice guidelines should be developed for improved medical practice in the prescription of medications for such a vulnerable population.

## Introduction

Polypharmacy is a common discussion subject in the peer-reviewed literature because it has become a major concern in the elderly. The elderly is the most susceptible to the negative effects of polypharmacy because of their altered pharmacokinetics and decreased drug clearance [1]. Such alterations coupled with the consumption of multiple medications could augment the risk of inappropriate drug utilization, drug-drug interactions, and adverse drug events [2–4]. Polypharmacy as well could play a major role in the development of frailty among older adults [5]. Studies have shown significant associations between polypharmacy in the elderly and undernourishment, impaired mobility, falls, nursing home placement, and hospitalization [6,7]. Polypharmacy could also lead to "prescription cascade" which occurs when drug-related side effects are misinterpreted as symptoms of a new disease or condition with consequent prescription of new medications. This may result in a chain of further adverse drug events and misdiagnoses [8]. Additionally, polypharmacy can increase the risk of mortality among this vulnerable population [9].

Several studies have reported that polypharmacy is associated with certain risk factors such as increasing age [10–13], female gender [11–13], Body Mass Index (BMI) [12, 14], and the number of co-morbidities [10, 13, 14]. Other studies have also shown that polypharmacy is associated with certain NCDs such as diabetes mellitus, hypertension, cardiovascular diseases, asthma, and dyslipidaemia [13–15].

Studies from different countries have reported varying rates polypharmacy in the elderly ranging from 18.0% in Brazil, 44.0% in Sweden, and 86.0% in South Korea [16–18]. In the United States, polypharmacy has tripled over two decades to reach 39% [19]. One primary reason for this variation may be that there is no clear universal definition for this phenomenon. Studies have used different definitions. A recent systematic review of the definitions of polypharmacy identified several definitions of this phenomenon, including numerical only (i.e., the number of medications); numerical in association with duration of therapy or healthcare setting; or a brief descriptive definition. The review concluded that the numerical definition, that is, the concurrent use of five or more medications daily, was the most frequently reported definition of polypharmacy [20].

A related phenomenon to the high prevalence rates in polypharmacy is that the elderly population is increasing globally [21]. The elderly population in the State of Qatar is also increasing due to public health initiatives and improved healthcare services across the country [22]. Associated with aging is the increasing prevalence of non-communicable diseases (NCDs) which necessitate the need for medication therapy [23, 24]. Therefore, investigating the prevalence of polypharmacy and its association with NCDs is crucial to implement measures that promote the rational use of medication. Thus, the objective of this study was to determine the

prevalence of polypharmacy ($\geq$ 5 drugs) and its association with NCDs in elderly ($\geq$65 years) Qatari patients attending Primary Healthcare (PHC) centers in Qatar.

## Methods

### Design and data source

A retrospective cross-sectional analysis was conducted using the Electronic Medical Record (EMR) database of the Primary Health Care Corporation (PHCC). PHCC is the largest provider of primary care in Qatar. This nonprofit organization delivers its services through 23 PHC centers distributed across the country according to population densities [25]. PHC centers are the most common first-line contact between community individuals and Qatar's healthcare system. Healthcare services provided in these health centers are free of charge to all residents of Qatar. Each of the 23 PHC centers serve a population made up of diverse demographic backgrounds such as ethnicity, education, income, and employment representative of Qatar's population [26]. According to the strategic plan of PHCC for 2019–2023, PHC centers aim at healthy aging in serving their population [27]. At the time of the study, patients attending PHC centers were usually seen by different primary physicians at different visits. Since the completion of our study, PHC centers adopted the family medicine model whereby patients are seen by their family medicine physician at each visit [27]. The EMR is the common documentation system for all healthcare professionals who provide direct services to all patients attending PHC centers.

The data retrieved from the EMR included patients' demographic and clinical characteristics such as age, gender, height, weight, and systolic and diastolic blood pressures. Also, the most frequent NCDs in this population were selected. These conditions included the following: diabetes, asthma, dyslipidaemia, hypertension, gastrointestinal reflux disease (GERD), cardiovascular diseases (ischemic heart disease, heart failure, arrhythmia, and stroke), arthritis (osteoarthritis and rheumatoid arthritis), and mental health conditions (depression, anxiety and dementia). Additionally, patients' prescribed medications were also recorded.

### Study population

The study population included all Qatari elderly patients ($\geq$65 years) who attended PHC centers during a period of six months from April 1 to September 30, 2017, and who had medication reconciliation done. According to PHCC policy, all patients attending PHC centers must have medication reconciliation done. This policy was introduced in January 2017 as a quality assurance measure because patients were seen by any available physicians at the time of their visit. Medication reconciliation is the process of identifying and listing patients' most current prescribed medications in comparison with all the medications that physicians have prescribed throughout the course of their treatment [28]. This process is performed by the multidisciplinary team including the patient's primary physician, nurse, and pharmacist. Their function is to ensure that the medication list is appropriately reviewed and verified (at different stages of patient encounter) to reduce possible medication errors such as duplicate medication orders or over/under doses.

### Measures

In this study, we defined polypharmacy as the concurrent use of five or more medications [20]. All medications were classified according to the 5th Level of the WHO's Anatomical Therapeutic Chemical (ATC) classification system [29]. We excluded drugs that were less likely to cause Drug Related Problems (DRP), such as dermatological (ATC-class D) and

topical products (ATC-class M02). Body mass index (BMI) was defined as weight/height$^2$. Overweight was defined as $25 \leq BMI < 30$ kg/m$^2$ and obesity as $BMI \geq 30$ kg/m$^2$. Clinical conditions were coded according to the International Classifications of Diseases, 10th revision (ICD-10).

## Data analysis

Descriptive statistics such as frequency tables, percentages, means and standard deviations (SD) were done for all variables of interest. Analytic statistics were applied as appropriate; Pearson's chi-square test was used to examine the factors associated with polypharmacy. Moreover, multivariate logistic regression analysis was performed to identify independent predictors of polypharmacy. Variables that had shown a significant association in the bivariate analysis (p-value $\leq 0.1$) were selected for the multivariate model. Odds ratios (ORs) and their 95% confidence intervals (CI) were calculated. Statistical significance was set at $p \leq 0.05$. All data were analyzed using IBM SPSS Statistics for Windows, version 23.0 (IBM Corp., Armonk, N.Y., USA).

## Ethics

The study protocol was approved by the PHCC—Independent Ethics Committee (PHCC-IEC) under reference number (PHCC/RS/17/11/015). The requirement for informed consents for patients was waved because all data from patients' Electronic Medical Record (EM) were accessed and analyzed anonymously. Confidentiality of the study participants was secured by anonymity of collected data. Moreover, the data were stored in a password-protected computer that was accessed only by the lead investigator.

## Results

As shown in Table 1, a total of 5639 patients were included in the study. The mean age of the study population was 72.8 (SD ± 6.5) years. Over 53% of the study population was female. The majority of the patients (84.8%) were either overweight or obese with a mean BMI of 31.1 (SD ± 6.3). Regarding NCDs, more than half of the study population (56.5%) had three or more NCDs and the mean number per person was 2.62 (SD ± 1.14). In addition, hypertension was the most frequent clinical diagnosis and was observed in more than three-quarter of the total sample (81.8%) followed by diabetes mellitus (74.1%).

Table 2 shows the comparison of NCDs between males and females. As shown, females had higher prevalence of NCDs than their male counterparts except for CVD. In addition, the proportion of those who had "three or more NCDs" was higher among females in comparison to males (60.1% vs. 52.3%; p-value <0.001). Also, obesity was more prevalent among females.

Table 3 shows the 20 medications most often prescribed for the elderly patients in the polypharmacy group and their distribution according to gender. Among these 20 medications, nine (45%) acted on the alimentary tract and metabolism (Group A). The second most frequent class was of medications acting on the cardiovascular system (Groups B and C).

### Prevalence of polypharmacy and its associated factors

In this study, 75.5% of the sample (95% CI: 74.3–76.6) were exposed to polypharmacy. The mean number of drugs prescribed for patients in the polypharmacy group was 9.78 (SD ± 4.00), while the mean number of drugs prescribed for patients in the non-polypharmacy group was 2.64 (SD ± 1.2).

**Table 1. Background characteristics of the study population (N = 5639).**

| Variables | Number (%) |
|---|---|
| Age (years) | |
| 65–69 | 2200 (39.0) |
| 70–74 | 1490 (26.4) |
| 75 or more | 1949 (34.6) |
| Gender | |
| Female | 3035 (53.8) |
| Male | 2604 (46.2) |
| Body Mass Index (kg/m$^2$) | |
| Underweight (<18.5) | 90 (02.0) |
| Normal (18.5–24.9) | 605 (13.2) |
| Overweight (25–29.9) | 1441 (31.5) |
| Obese (≥30) | 2441 (53.3) |
| No. of chronic conditions | |
| Zero | 244 (04.3) |
| One | 744 (13.2) |
| Two | 1461 (25.9) |
| Three | 1611 (28.6) |
| Four or more | 1572 (27.9) |
| Hypertension | |
| Yes | 4651 (81.8) |
| No | 1023 (18.2) |
| Diabetes mellitus | |
| Yes | 4176 (74.1) |
| No | 1463 (25.9) |
| Dyslipidemia | |
| Yes | 2445 (43.4) |
| No | 3194 (56.6) |
| Cardiovascular disease [a] | |
| Yes | 1069 (20.7) |
| No | 4470 (79.3) |
| Asthma | |
| Yes | 1152 (20.4) |
| No | 4486 (79.6) |
| Arthritis [b] | |
| Yes | 1033 (18.3) |
| No | 4606 (81.7) |
| Gastroesophageal reflux disease | |
| Yes | 708 (12.6) |
| No | 4931 (87.4) |
| Mental health conditions [c] | |
| Yes | 177 (3.1) |
| No | 5462 (96.9) |

[a] Ischemic heart disease, heart failure, arrhythmia, and stroke

[b] osteoarthritis and rheumatoid arthritis

[c] depression, anxiety and dementia

**Table 2. Comparison of non-communicable diseases between males and females (N = 5639).**

| Variable | Males | Females | p-value |
|---|---|---|---|
| | Count (%) | Count (%) | |
| No. of chronic conditions | | | <0.001* |
| Zero | 131 (5.0) | 113 (3.7) | |
| One | 390 (15.0) | 354 (11.7) | |
| Two | 719 (27.6) | 742 (24.5) | |
| Three or more | 1361 (52.3) | 1821 (60.1) | |
| Body Mass Index (kg/m$^2$) | | | <0.001* |
| Underweight (<18.5) | 57 (2.2) | 33 (1.1) | |
| Normal (18.5–24.9) | 411 (15.8) | 194 (6.4) | |
| Overweight (25–29.9) | 867 (33.3) | 574 (18.9) | |
| Obese (≥30) | 784 (30.1) | 1657 (54.6) | |
| Hypertension | | | 0.001* |
| Yes | 2085 (80.1) | 2530 (83.4) | |
| No | 518 (19.9) | 505 (16.6) | |
| Diabetes mellitus | | | 0.017* |
| Yes | 1893 (72.7) | 2283 (75.2) | |
| No | 711 (27.3) | 752 (24.8) | |
| Dyslipidemia | | | <0.001* |
| Yes | 966 (37.1) | 1479 (48.7) | |
| No | 1638 (62.9) | 1556 (51.3) | |
| Cardiovascular disease | | | <0.001* |
| Yes | 664 (25.5) | 505 (16.6) | |
| No | 1940 (74.5) | 2530 (83.4) | |
| Asthma | | | <0.001* |
| Yes | 430 (16.5) | 722 (23.8) | |
| No | 2174 (83.5) | 2312 (76.2) | |
| Arthritis / Osteoarthritis | | | <0.001* |
| Yes | 394 (15.1) | 639 (21.1) | |
| No | 2210 (84.9) | 2396 (78.9) | |
| Gastroesophageal reflux disease | | | 0.001* |
| Yes | 287 (11.0) | 421 (13.9) | |
| No | 2317 (89.0) | 2614 (86.1) | |
| Mental disorders | | | <0.001* |
| Yes | 57 (2.2) | 120 (4.0) | |
| No | 2547 (97.8) | 2915 (96.0) | |

As shown in Table 4, the bivariate analysis showed no statistically significant difference in the prevalence of polypharmacy between different age groups. However, females had a higher prevalence of polypharmacy as compared to males (77.5% vs. 73.2%, p<0.001). Moreover, the prevalence of polypharmacy was significantly higher in patients with increasing BMI. Similarly, polypharmacy was higher among patients who had a higher number of NCDs. As well, polypharmacy was higher among patients with hypertension (80.0% vs. 55.0%, p<0.001), diabetes mellitus (82.4% vs. 55.9%, p<0.001), dyslipidemia (82.1% vs. 70.4%, p<0.001), cardiovascular disease (86.1% vs. 72.7%, p<0.001), asthma (82.9% vs. 73.6%, p<0.001), GERD (79.1% vs. 75.0%, p<0.01) and mental disorders (82.5% vs. 75.3%, p<0.05).

The multivariate logistic regression analysis is shown in Table 5. Females were 1.18 times more likely to have polypharmacy compared to males (95% CI: 1.03–1.34). Chronic conditions

**Table 3. Anatomical Therapeutic Chemical (ATC) Level 5 drug classes most frequently used by subjects exposed to polypharmacy and their distribution according to gender.**

| Drug name | ATC* class (Level 5) | Total | Female | Male |
|---|---|---|---|---|
| | | (N = 4257) (%) | (n = 2352) (%) | (n = 1905) (%) |
| Ergocalciferol | A11CC01 | 67.8 | 75.1 | 58.9 |
| Acetylsalicylic acid | B01AC06 | 56.1 | 52.4 | 60.7 |
| Atorvastatin | C10AA05 | 42.1 | 40.1 | 44.6 |
| Metformin | A10BA02 | 36.0 | 37.9 | 33.7 |
| Rosuvastatin | C10AA07 | 27.5 | 29.3 | 25.4 |
| Amlodipine | C08CA01 | 23.6 | 23.6 | 23.7 |
| Gliclazide | A10BB09 | 23.3 | 22.4 | 24.3 |
| Pantoprazole | A02BC02 | 23.0 | 22.7 | 23.3 |
| Sitagliptin-metformin | A10BD07 | 22.1 | 21.3 | 23.0 |
| Paracetamol-orphenadrine | M03BC51 | 21.0 | 24.2 | 17.0 |
| Rabeprazole | A02BC04 | 19.5 | 21.9 | 16.5 |
| Paracetamol | N02BE01 | 18.5 | 21.3 | 15.0 |
| Esomeprazole | A02BC05 | 16.9 | 17.5 | 16.1 |
| Levothyroxine sodium | H03AA01 | 16.4 | 22.5 | 8.9 |
| Salbutamol | R03AC02 | 16.3 | 18.0 | 14.1 |
| Insulin glargine | A10AE04 | 16.0 | 16.4 | 15.6 |
| Celecoxib | M01AH01 | 12.9 | 13.9 | 11.6 |
| Sitagliptin | A10BH01 | 12.6 | 12.6 | 12.7 |
| Furosemide | C03CA01 | 12.5 | 10.6 | 14.8 |
| Clopidogrel | B01AC04 | 12.2 | 8.2 | 17.1 |

* ACT = Anatomical Therapeutic Chemical Classification.

of hypertension, diabetes, dyslipidemia, cardiovascular disease and asthma were found to be significant independent parameters associated with polypharmacy. In addition, the BMI and number of NCDs were independently associated with polypharmacy.

## Discussion

The present study examined the prevalence of polypharmacy and its association with NCDs in Qatari elderly patients attending PHC centers by accessing the EMR database of the 23 PHC centers. Our study showed that almost three-quarters of the study population were exposed to polypharmacy (≥5 drugs). Also, our findings confirmed that female gender, BMI, and the number of certain NCDs to be significantly associated with polypharmacy.

The prevalence of polypharmacy in the present study is dramatically higher than that identified in other regional and international studies. A recent study conducted in Saudi Arabia among a sample of 3009 older adults found that 55% of them were exposed to polypharmacy [30]. Also, a study in Kuwait showed that more than 60% of the older adults had polypharmacy [31]. In addition, a study in Italy reported that the prevalence of polypharmacy among a national sample of elderly individuals was 40% [10], while a study in Sweden was 44.0% [17]. Another study in the United States showed that the prevalence of polypharmacy among older adults was 39% [19]. Similarly, a study from Brazil of 1705 elderly individuals reported that 32% were exposed to polypharmacy [11].

On the other hand, some studies reported higher prevalence of polypharmacy than that found in our study. For example, a population-based study in Korea found that 86.4% of elderly patients had polypharmacy [18]. Furthermore, a study in Oman in 2016 reported that

**Table 4. Bivariate association between explanatory variables and polypharmacy status (N = 5639).**

| Variable | No polypharmacy | Polypharmacy | χ2 test | p-value |
|---|---|---|---|---|
| | Number (%) | Number (%) | | |
| Age (years) | | | 3.76 | 0.152 |
| 65–69 | 565 (25.7) | 1635 (74.3) | | |
| 70–74 | 341 (22.9) | 1149 (77.1) | | |
| 75 or more | 476 (24.4) | 1473 (75.6) | | |
| Gender | | | 14.26 | <0.001* |
| Male | 699 (26.8) | 1905 (73.2) | | |
| Female | 683 (22.5) | 2352 (77.5) | | |
| Body Mass Index (kg/m$^2$) | | | 79.47 | <0.001* |
| Underweight (<18.5) | 27 (30.0) | 63 (70.0) | | |
| Normal (18.5–24.9) | 151 (25.0) | 454 (75.0) | | |
| Overweight (25–29.9) | 339 (23.5) | 1102 (76.5) | | |
| Obese (≥30) | 500 (20.5) | 1941 (79.5) | | |
| No. of chronic conditions | | | 584.61 | <0.001* |
| Zero | 157 (64.3) | 87 (35.7) | | |
| One | 351 (47.2) | 393 (52.8) | | |
| Two | 398 (27.2) | 1063 (72.8) | | |
| Three | 268 (16.6) | 1342 (83.4) | | |
| Four or more | 207 (13.2) | 1365 (86.8) | | |
| Hypertension | | | 282.55 | <0.001* |
| No | 460 (45.0) | 563 (55.0) | | |
| Yes | 922 (20.0) | 3693 (80.0) | | |
| Diabetes mellitus | | | 409.34 | <0.001* |
| No | 645 (44.1) | 818 (55.9) | | |
| Yes | 737 (17.6) | 3439 (82.4) | | |
| Dyslipidemia | | | 102.70 | <0.001* |
| No | 945 (29.6) | 2249 (70.4) | | |
| Yes | 437 (17.9) | 2008 (82.1) | | |
| Cardiovascular disease [a] | | | 88.95 | <0.001* |
| No | 1219 (27.3) | 3251 (72.7) | | |
| Yes | 163 (13.9) | 1006 (86.1) | | |
| Asthma | | | 42.98 | <0.001* |
| No | 1185 (26.4) | 3301 (73.6) | | |
| Yes | 197 (17.1) | 955 (82.9) | | |
| Arthritis [b] | | | 2.11 | 0.078 |
| No | 1147 (24.9) | 3459 (75.1) | | |
| Yes | 235 (22.7) | 798 (77.3) | | |
| Gastroesophageal reflux disease | | | 5.68 | 0.009* |
| No | 1234 (25.0) | 3697 (75.0) | | |
| Yes | 148 (20.9) | 560 (79.1) | | |
| Mental health conditions [c] | | | 4.83 | 0.015* |
| No | 1351 (24.7) | 4111 (75.3) | | |
| Yes | 31 (17.5) | 146 (82.5) | | |

[a] Ischemic heart disease, heart failure, arrhythmia, and stroke

[b] osteoarthritis and rheumatoid arthritis

[c] depression, anxiety and dementia

**Table 5. Multivariate logistic regression analysis of predictors of polypharmacy.**

| Variable | Adjusted OR (95% CI) | p-value |
|---|---|---|
| Age (years) | | 0.32 |
| 65–69 | 1 | |
| 70–74 | 1.13 (0.96–1.34) | |
| 75 or more | 0.54 (0.89–1.22) | |
| Gender | | 0.013* |
| Male | 1 | |
| Female | 1.18 (1.03–1.34) | |
| Body Mass Index (kg/m²) | | <0.001* |
| Underweight (<18.5) | 1 | |
| Normal (18.5–24.9) | 1.28 (0.79–2.09) | |
| Overweight (25–29.9) | 1.39 (0.87–2.22) | |
| Obese (≥30) | 1.66 (1.04–2.63) | |
| No. of chronic conditions | | 0.01* |
| Zero | 1 | |
| One | 1.14 (0.82–1.60) | |
| Two | 1.54 (1.03–2.31) | |
| Three | 1.91 (1.17–3.12) | |
| Four or more | 1.76 (1.00–3.14) | |
| Hypertension | | <0.001* |
| No | 1 | |
| Yes | 1.71 (1.38–2.13) | |
| Diabetes mellitus | | <0.001* |
| No | 1 | |
| Yes | 2.38 (1.97–2.87) | |
| Dyslipidemia | | 0.008* |
| No | 1 | |
| Yes | 1.29 (1.06–1.56) | |
| Cardiovascular disease [a] | | <0.001* |
| No | 1 | |
| Yes | 1.56 (1.25–1.95) | |
| Asthma | | <0.001* |
| No | 1 | |
| Yes | 1.39 (1.13–1.72) | |
| Gastroesophageal reflux disease | | 0.80 |
| No | 1 | |
| Yes | 0.96 (0.75–1.24) | |
| Mental health conditions [c] | | 0.33 |
| No | 1 | |
| Yes | 1.22 (0.79–1.88) | |

OR = Odds ratio; CI = Confidence intervals

*statistically significant = $p < 0.05$

[a] Ischemic heart disease, heart failure, arrhythmia, and stroke

[b] osteoarthritis and rheumatoid arthritis

[c] depression, anxiety and dementia

76.3% of elderly patients discharged from a tertiary hospital were exposed to polypharmacy [32]. However, such findings in Oman may be explained by the additional number of

medications that were prescribed for the acute conditions of these patients. A possible explanation for the high prevalence of polypharmacy found in our study may be that in PHC centers the patients are seen and treated by different physicians resulting in the prescription of additional medications at different visits. A study by Kann et al. (2015) reported that the risk of polypharmacy in patients increases significantly with the number of prescribers (OR: 2.32; 95% CI: 2.31–2.33) [33]. One other factor that could explain such a phenomenon is the high prevalence of NCDs in our study population. Furthermore, a study in Canada showed a significant association between polypharmacy and higher frequency of family physician visits in elderly patients [34]. Finally, the provision of medications at no cost to all Qatari citizens might make it easier for physicians to prescribe them.

The list of the 20 most used medications among those exposed to polypharmacy (Table 3) reflects the most prevalent NCDs. Six medications on the list were medications acting on the cardiovascular system (hypertension, dyslipidemia and CVD). Five medications were for treating diabetes mellitus. Three medications on this list (pantoprazole, rabeprazole, and esomeprazole) reduce gastric acid and are indicated for the treatment of conditions such as GERD and peptic ulcers. These are in line with other studies that have reported associations between polypharmacy and common comorbidities in the elderly [10, 11, 13, 31, 32].

Consistent with other studies the prevalence of polypharmacy in the present study was significantly higher among females [11–13, 15, 16]. In contrast, other studies have reported higher polypharmacy rates in males [10, 18]. Such inconsistencies among study findings could be explained by differences in physicians' prescribing approaches toward genders as well as to differences between genders and their health-seeking behaviors. Consistent with another study in Qatar [23], the prevalence of almost all of the NCDs we studied, except for cardiovascular diseases, was higher in females than in males.

Our results demonstrated that the rate of polypharmacy was positively associated with BMI. Similar findings have also been reported by Carmona-Torres (2018) in Spain [12], Slater et al. (2018) in the United Kingdom [14], and Ramos et al. (2016) in Brazil [16]. This can be attributed to the fact that overweight/obesity is considered a risk factor for several NCDs, which in turn may require a higher number of medications for treatment.

Finally, in our study, polypharmacy was substantially increased with the increasing number of NCDs. This is consistent with findings from other studies [10, 14, 15, 18, 19]. Also, polypharmacy showed a stronger association with certain NCDs than others. For instance, the logistic regression model in our study confirmed the solid relationship between polypharmacy and the following NCDs: diabetes mellitus, hypertension, cardiovascular diseases, asthma, and dyslipidaemia. These findings are in alignment with the results of other studies [12, 13, 15–18].

## Study strengths and limitations

The present study is the first to examine the prevalence of polypharmacy and its associated factors among elderly citizens in Qatar. The strength of our study lies in its large sample size, which allowed for statistical analysis with sufficient statistical power. Therefore, the findings from this study provide a reliable basis to confirm that high polypharmacy exists in primary healthcare settings in Qatar. Moreover, the use of standardized classification systems such as the ATC classification made the results more valid and reliable and enabled comparison with other studies.

On the other hand, the lack of a standard definition of polypharmacy across studies made comparisons difficult. Moreover, the calculated prevalence might be overestimated because patients who visited the PHC centers might have suffered from multiple health conditions. Another limitation is that some of the variables (e.g., socio-economic status, marital status, or

the number of prescribers) was not consistently recorded for most patients making us unable to include them in our analysis. One other limitation in our study is that we relied on medications prescribed and reconciled, but not on adherence. It would have been valuable to know to what extent the patients in the study were adhering to their prescribed medication. Non-adherence to medical regimen has been recognized as major concern in medical practice particularly in patients with multi-morbidities [35]. Clinical guidelines may guide clinical decision making particularly in patients with NCDs and co-morbidities and associated polypharmacy [36]. Finally, because of our study design, our findings cannot be generalized to the entire population of interest, but can be only applied to the population included in the study.

## Conclusions

This study provided evidence that the prevalence of polypharmacy among Qatari elderly attending PHC centers is very high with almost three-quarters of the study population exposed to it. The study as well demonstrated a significant association between polypharmacy and BMI with about 80% of the study subjects being obese or overweight. Our findings confirmed the strong relationship between polypharmacy and NCDs such as hypertension, diabetes mellitus, dyslipidaemia, cardiovascular diseases and asthma. As appropriate care for elderly patients is increasingly challenging, targeted educational programs should be developed for healthcare professionals to raise their awareness of the magnitude and negative impact of polypharmacy. Furthermore, PHC centers should establish best practice guidelines for improved medical practice in the prescription of medications for such a vulnerable population.

## Supporting information

**S1 Data.**
(SAV)

## Author Contributions

**Conceptualization:** Ayman Al-Dahshan, Noora Al-Kubiasi, Wael Saeed.

**Data curation:** Ayman Al-Dahshan, Noora Al-Kubiasi, Wael Saeed.

**Formal analysis:** Ayman Al-Dahshan.

**Methodology:** Ayman Al-Dahshan.

**Supervision:** Iheb Bougmiza.

**Validation:** Manal Al-Zaidan.

**Writing – original draft:** Ayman Al-Dahshan.

**Writing – review & editing:** Manal Al-Zaidan, Vahe Kehyayan, Iheb Bougmiza.

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
