## [Decision Letter · Decision Letter 0]

14 Jan 2020

PONE-D-19-22287

Prevalence of polypharmacy and its associated factors among Qatari elderly patients attending primary healthcare centers, 2017

PLOS ONE

Dear Dr. Al Dahshan,

Thank you for submitting your manuscript to PLOS ONE. After careful consideration, we have decided that your manuscript does not meet our criteria for publication and must therefore be rejected.

Specifically: The paper does not add any value to the existing literature on polypharmacy. The authors should examine other important social and/or health-system-level factors that contribute to polypharmacy. Polypharmacy is not well defined in the paper. Additionally, the paper has a lot of grammatical errors and is not clear to read.  

ACADEMIC EDITOR: 

Here are some specific comments:

1. In the introduction section, the authors talk about physician-related factors that may contribute to polypharmacy, however, such factors are not included in the study.

2. How do you define simultaneous use of five or more medications per person?

3. The authors should also look at polypharmacy as a continuous measure.

4. Data analysis and results sections are not properly written and are not organized well. There are many grammatical mistakes.

Discussion:

*“**A possible explanation for the high prevalence of polypharmacy found in our study is the fact that elderly patients are being seen and treated by different health care professionals, which might result in adding new prescription, at different occasions.”*

If the authors think that continuity of care is an important factor associated with polypharmacy then why was it not included.

I am sorry that we cannot be more positive on this occasion, but hope that you appreciate the reasons for this decision.

Yours sincerely,

Satya Surbhi, PhD

Academic Editor

PLOS ONE

Reviewers' comments:

Reviewer's Responses to Questions

**Comments to the Author**

1. Is the manuscript technically sound, and do the data support the conclusions?

Reviewer #1: Partly

2. Has the statistical analysis been performed appropriately and rigorously? 

Reviewer #1: Yes

3. Have the authors made all data underlying the findings in their manuscript fully available?

Reviewer #1: Yes

4. Is the manuscript presented in an intelligible fashion and written in standard English?

Reviewer #1: Yes

5. Review Comments to the Author

Reviewer #1: Dear Authors,

Thank you for give me the opportunity to review this manuscript. It treats an interesting matter for health care workers. I hope my comments will be useful to do that.

The manuscript entitled “Prevalence of polypharmacy and its associated factors among Qatari elderly patients attending primary healthcare centers, 2017” is based on a study that was carried out with the purpose to investigate the prevalence and the associated factors of polypharmacy among the elderly (≥65 years) nationals in this population.

Given increased survival for older individuals, including those with multiple chronic conditions and polypharmacy, the topic has high significance and potential impact. Overall, the manuscript contributes to a better understanding of this issue.

You have performed an interesting work, where the study problem, as well as the different concepts, are clearly described. The backgrounds are exposed and the need to carry out the study is justified.

However, some concerns should be addressed. Please see the following questions:

Abstract

P.2 line 33, please, provide a definition of the abbreviation BMI.

Introduction section

P.5 line 97 Main objective

There is an inconsistency between the definition that the main objective is made in this section with the one made in the abstract. Given the small number of variables that are taken into account and described in the results, the title of the study can be misleading since the results focus primarily on the relationship between polypharmacy and the main comorbidities observed in the population of study. Therefore, it would be more appropriate for both the title and the objective of the study to refer to the relationship between polypharmacy and such comorbidities.

Methods section

P.6 line 116 Study population.

Reading this section I have raised many doubts. First, it is observed that the study / sample population is made up of the elderly patients (> 65 years old) registered in the Electronic Medical Record (EMR) and who had at least one visit to any primary care center during the study period . This can be a selection bias since the patients who go to the health centers may be the ones with the most pathologies and therefore the most polymedicated. In this case, the results could not be extrapolated to the general Qatari population over 65 years old. What happens to patients not registered in the EMR or who did not go to any health center during the study period? Are they different from those studied?

On the other hand, you talk about medication reconciliation but it is not explained what this procedure consists of as well as the functions of the different members of the multidisciplinary team that participated in it.

P6 line 122. Definition and Measures.

Polypharmacy is defined as the simultaneous use of 5 or more medications per patient. While this definition of polypharmacy is the most frequently described in the literature, as you explain in the introduction, however, this is a poor and very variable definition that would explain the high prevalence of polymedication that you have found in your research. If we want to take into account the possible negative effects that polymedication can have on the health of patients, a definition of polymedication that takes into account the time variable should be used.

Furthermore, if one of the objectives of the study is to investigate the associated factors of polypharmacy, it would be advisable to include more variables than those studied such as educational level, socio-economic level, marital status, place of residence, number of prescribers, number of visits to health services, etc. In addition, the section on material and methods should include a section where these variables are described and categorized, both dependent and independent variables. Otherwise both the objective and the title of the study should be changed.

P.7 line 149. Ethical considerations

There is no mention that the consent of patients has been obtained to be included in the study by signing an informed consent.

Results section

Table 1. Please provide a definition of the different categories of the Body Mass Index variable.

Study strengths and limitations section

The fact that the way in which the sample of the study has been selected may imply a selection bias should be included in this section, as well as the lack of convenience of generalizing the results obtained to the general Qatari population older than 65 years.

Conclusions section

The conclusions should be more concrete, referring exclusively to the proposed objectives and be supported by the results obtained. Recommendations and personal reflections that are not supported by concrete results should be included in another section of the manuscript such as discussion, prospective or recommendations.

6. PLOS authors have the option to publish the peer review history of their article (what does this mean?). If published, this will include your full peer review and any attached files.

Reviewer #1: No

- - - - -

---

## [Author Response · Author response to Decision Letter 0]

21 Mar 2020

PONE-D-19-22287 

Prevalence of polypharmacy and its associated factors among Qatari elderly patients ‎attending primary healthcare centers, 2017 

RESPONSE TO REVIEWERS

General Response: ‎

We appreciated the many insightful comments made by the reviewers. We studied them ‎carefully and made diligent efforts in revising our manuscript. As a result, we consider our ‎manuscript much improved to the reviewers’ satisfaction.

 ‎

Academic Editor’s Comments:‎

Comment #1:‎

In the introduction section, the authors talk about physician-related factors that may ‎contribute to polypharmacy, however, such factors are not included in the study.‎

Response:‎

Thank you for bringing this issue to our attention. This was not our intent and we can ‎appreciate the reviewer’s concern. Our focus was to study the prevalence of polypharmacy ‎and its association with major non-communicable diseases and other factors (e.g.: gender, ‎age, BMI, etc.) You will note in the revised manuscript that we have removed the reference ‎to “physician-related factors”. ‎

Comment #2:‎

How do you define simultaneous use of five or more medications per person?‎

Response:‎

Thank you for your comment. By simultaneous use of five or more medications per person” ‎we meant concurrent use of five or more medications. That is, we characterized ‎polypharmacy as “5 or more medications”. This point was addressed in the revised ‎manuscript. (Methods section: P. 5, line 103-104)‎

Comment #3:‎

The authors should also look at polypharmacy as a continuous measure.‎

Response:‎

Thank you for your suggestion about treating polypharmacy as a continuous variable. We ‎agree that this would be an appropriate option. However, the reason we chose to treat it as a ‎categorical variable was (1) we found in our literature review that most studies were treating ‎it as categorical, and (2) we wanted to be able to compare our findings with others. ‎

Comment #4:‎

Data analysis and results sections are not properly written and are not organized well. There ‎are many grammatical mistakes. ‎

Response:‎

Thank you for this feedback. We have engaged the services of a native English speaker to ‎review the entire manuscript and make the necessary corrections. The data analysis section ‎was modified as advised. (Method section: P. 6, line 111-118)‎

Comment #5:‎

Discussion: “A possible explanation for the high prevalence of polypharmacy found in our ‎study is the fact that elderly patients are being seen and treated by different health care ‎professionals, which might result in adding new prescription, at different occasions.”‎

If the authors think that continuity of care is an important factor associated with ‎polypharmacy then why was it not included.‎

Response:‎

Thank you for your observation. At the time of our study, patients attending Qatar’s primary ‎health centers were usually seen by different primary physicians at different visits. However, ‎the family medicine model was recently implemented across all primary health centers in the ‎country and consequently patients are now being seen by their own family medicine ‎physician at each visit. In the discussion section of the manuscript we have added text about ‎the issue of lack of continuity of care. (Discussion section: P. 12, line 187-191)‎

Reviewer’s Comments:‎

Dear Authors,‎

‎“Thank you for give me the opportunity to review this manuscript. It treats an interesting ‎matter for health care workers. I hope my comments will be useful to do that.‎

The manuscript entitled “Prevalence of polypharmacy and its associated factors among Qatari ‎elderly patients attending primary health centers, 2017” is based on a study that was carried ‎out with the purpose to investigate the prevalence and the associated factors of ‎polypharmacy among the elderly (≥65 years) nationals in this population.‎

Given increased survival for older individuals, including those with multiple chronic ‎conditions and polypharmacy, the topic has high significance and potential impact. Overall, ‎the manuscript contributes to a better understanding of this issue.‎

You have performed an interesting work, where the study problem, as well as the different ‎concepts, are clearly described. The backgrounds are exposed and the need to carry out the ‎study is justified. However, some concerns should be addressed. Please see the following ‎questions:”‎

Response: Thank you for this positive comment. ‎

Comment #1:‎

Abstract: P.2 line 33: “Please, provide a definition of the abbreviation BMI”.‎

Response:‎

A definition of BMI was added (BMI= Body Mass Index). (Abstract: P. 2, Line 32-33)‎

Comment #2:‎

Introduction section: P.5, line 97 Main objective: “There is an inconsistency between the ‎definition that the main objective is made in this section with the one made in the abstract”.‎

Response:‎

Thank you for bringing this discrepancy to our attention. We have corrected this discrepancy. ‎The objective now reads “to determine the prevalence of polypharmacy (≥ 5 drugs) and its ‎association with non-communicable diseases (NCDs) in elderly (≥65 years) Qatari patients ‎attending Primary Healthcare (PHC) centers in Qatar”. (Abstract: P.2, line 22-24), (Intro. ‎section: P.4, line 74-76)‎

Comment #3:‎

‎“Given the small number of variables that are taken into account and described in the results, ‎the title of the study can be misleading since the results focus primarily on the relationship ‎between polypharmacy and the main comorbidities observed in the population of study. ‎Therefore, it would be more appropriate for both the title and the objective of the study to ‎refer to the relationship between polypharmacy and such comorbidities”.‎

Response:‎

The title was modified so it reflects the study objective as follows: (Prevalence of ‎polypharmacy and the association with non-communicable diseases in Qatari ‎elderly patients ‎attending primary healthcare centers: A cross-sectional study‎). (Title page: P. 1, line 2-3)‎

Comment #4:‎

Methods section: P.6 line 116 Study population:‎

‎“Reading this section I have raised many doubts. First, it is observed that the study / sample ‎population is made up of the elderly patients (> 65 years old) registered in the Electronic ‎Medical Record (EMR) and who had at least one visit to any primary health center during the ‎study period . This can be a selection bias since the patients who go to the health centers ‎may be the ones with the most pathologies and therefore the most polymedicated. In this ‎case, the results could not be extrapolated to the general Qatari population over 65 years ‎old. What happens to patients not registered in the EMR or who did not go to any health ‎center during the study period? Are they different from those studied?”‎

Response:‎

The Primary Health Care Corporation is the largest provider of primary care in the State of ‎Qatar. This non-profit organization ‎delivers its services through 23 primary health centers ‎distributed over the country according to population ‎density. Furthermore, PHCs are the ‎common and popular first-line level of contact between community individuals and ‎their ‎healthcare system. The registered number elderly nationals in EMR at the time of the study ‎was 15,286 while the total number of the same population in Qatar in 2017 was 17,895 ‎‎(according to the planning and statistics authority in Qatar). Thus, this indicates that more ‎than 85% of Qatari elderly in the country were registered in the EMR.‎

Overall, findings from this study provide a reliable basis for polypharmacy phenomenon in ‎‎primary healthcare settings. Therefore, I do agree with the reviewer comment that the study ‎findings could not be generalised to the whole elderly Qatari population, but to Qatari elderly ‎patients attending PHC centers. This point was clarified in the study strengths and limitation ‎section of the manuscript. (Study strengths and limitations: P. 14, line 221-223; 226-‎‎227; 230-231).‎

Comment #5:‎

‎“On the other hand, you talk about medication reconciliation but it is not explained what this ‎procedure consists of as well as the functions of the different members of the ‎multidisciplinary team that participated in it”.‎

Response:‎

Thank you for this comment. Medication reconciliation is the process of identifying and listing ‎of patients’ most current medications that they are taking in comparison with all the ‎medications that their physicians have prescribed throughout the course of the patients’ ‎treatment. (1)‎

The multidisciplinary team‎ involved in medication reconciliation includes the primary care ‎physician, the nurse and the pharmacist. Their function is to ensure that the medication list is ‎appropriately reviewed and verified (at different stages of patient encounter) to reduce ‎possible medication errors such as duplicate medication orders or over/under doses. ‎This ‎point was clarified in the manuscript. (Methods: P. 5, line 95-101).‎

‎1. Institute for Healthcare Improvement. Medical Reconciliation to Prevent Drug Events. 2020; retrieved from ‎‎[http://www.ihi.org/Topics/ADEsMedicationReconciliation/Pages/default.aspx]‎

Comment #6:‎

Definition and Measures: P6 line 122.‎

‎“Polypharmacy is defined as the simultaneous use of 5 or more medications per patient. ‎While this definition of polypharmacy is the most frequently described in the literature, as ‎you explain in the introduction, however, this is a poor and very variable definition that ‎would explain the high prevalence of polymedication that you have found in your research. If ‎we want to take into account the possible negative effects that polymedication can have on ‎the health of patients, a definition of polymedication that takes into account the time ‎variable should be used”.‎

Response:‎

We appreciate this comment. However, we based our definition of polypharmacy on a 2017 ‎systematic review of 111 studies that was conducted to identify and summarize ‎‎polypharmacy definitions in existing literature (2). The most commonly reported definition of ‎‎polypharmacy (80.4%) was the numerical definition of five or more medications daily; ‎moreover, only ‎‎10% of all included studies used numerical definitions of polypharmacy which ‎also included a ‎duration of treatment in the definition. ‎

In our study, it was not feasible to obtain information ‎on the duration of treatment. As a ‎result, we decided to use the numerical definition of five ‎or more medications daily‎. Also, ‎using such a definition, we were able to compare our findings to many others who used the ‎same definition. To further illustrate this, we have now constructed a new table that will show ‎the most frequently (top 20) used medications by patients exposed to polypharmacy. The ‎table demonstrates that most prescriptions were aimed to treat chronic conditions ‎‎(medications for long-term use). (Results: P. 8, line 136-141); (Discussion: P. 13, line 195-‎‎201)‎

‎2. Masnoon N, Shakib S, Kalisch-Ellett L, Caughey G. What is polypharmacy? A systematic review of definitions. ‎BMC Geriatr. 2017;17(1).‎

Comment #7: ‎

‎“Furthermore, if one of the objectives of the study is to investigate the associated factors of ‎polypharmacy, it would be advisable to include more variables than those studied such as ‎educational level, socio-economic level, marital status, place of residence, number of ‎prescribers, number of visits to health services, etc. In addition, the section on material and ‎methods should include a section where these variables are described and categorized, both ‎dependent and independent variables. Otherwise both the objective and the title of the ‎study should be changed”.‎

Response:‎

Thank you for this insightful comment. We recognize that this is a limitation of our study. ‎Considering that the data were obtained from the EMR, some variables were not consistently ‎recorded for most patients and thus we were unable to include them in the analysis. (Study ‎strengths and limitations: P. 14, line 228-230) ‎

Also, both the objective and the title of the study were modified as advised. (please see our ‎responses to comments #2 and 3) ‎

Comment #8:‎

Ethical considerations: P.7 line 149.‎

‎“There is no mention that the consent of patients has been obtained to be included in the ‎study by signing an informed consent”.‎

Response:‎

Thank you for this comment. This study was a retrospective study using data from patients’ ‎electronic medical record. All data were anonymized. The Primary Health Care Corporation-‎Independent Ethics Committee (PHCC-IEC) waived the requirement to obtain any informed ‎consent since the data used in this study were anonymized (‎reference number ‎PHCC/RS/17/11/015). ‎ This point was clarified in the manuscript. (Methods: P. 6, line 120-‎‎124).‎

Comment #9:‎

Results section:‎

Table 1. Please provide a definition of the different categories of the Body Mass Index ‎variable”.‎

Response:‎

A definition for each BMI category was added as follows: Underweight (<18.5)‎, Normal (18.5-‎‎24.9)‎, Overweight (25-29.9)‎, Obese (≥30)‎. (Results: Table 1, P. 7)‎

Comment #10:‎

Study strengths and limitations section

‎“The fact that the way in which the sample of the study has been selected may imply a ‎selection bias should be included in this section, as well as the lack of convenience of ‎generalizing the results obtained to the general Qatari population older than 65 years”.‎

Response:‎

We agree with your comment. (Please see our response to comment #4)

 ‎

Comment #11:‎

Conclusions section

‎“The conclusions should be more concrete, referring exclusively to the proposed objectives ‎and be supported by the results obtained. Recommendations and personal reflections that ‎are not supported by concrete results should be included in another section of the ‎manuscript such as discussion, prospective or recommendations.”‎

Response:‎

Thank you for this insightful observation. The conclusion section was revised as suggested. ‎‎(Abstract: P. 2, line 35-39); (Conclusions section: P. 14-15, line 233-242)‎

‎ ‎

---

## [Editor Report · Decision Letter 1]

4 May 2020

PONE-D-19-22287R1

Prevalence of polypharmacy and the association with non-communicable diseases in Qatari elderly patients attending primary healthcare centers: A cross-sectional study

PLOS ONE

Dear Dr. Al-Dahshan,

Thank you for submitting your manuscript to PLOS ONE. After careful consideration, we feel that it has merit but does not fully meet PLOS ONE’s publication criteria as it currently stands. Therefore, we invite you to submit a revised version of the manuscript that addresses the points raised during the review process.

ACADEMIC EDITORS: 

We have concluded that a major revision should be performed for the paper to be published according to Plos One criteria. Moreover a bit more speculation might be useful to suggest strategies for clinical improvement or further investigations.

The main point is represented by external validity. It seems clear that the data are not representative of general elderly population in Qatar: prevalence of obesity >50%, diabetes almost 80%, asthma 20% are largely greater than population estimates around the world, and may clearly explain the high prevalence of polypharmacy – almost double in comparison with international estimates (to be cited): 39% in US according to ref 18, 44% in Sweden according to Morin L et al., Clin Epidemiol, 2018). In fact Authors state: "The study population included all Qatari elderly patients (≥65 years) who attended PHC centers during a period of six months from April 1 to September 30, 2017, AND WHO HAD MEDICATION RECONCILIATION DONE".

What proportion of older Qataris attend PHC? And, most important, which percentage of older patients who attended the PHC underwent a medication reconciliation? Is the sample at least representative of older subjects who attend a Primary Health Center? If this is the case, conclusions should be moderated, stating that the prevalence of polypharmacy of older subjects ATTENDING A PHC in Qatar is very high, and discussion may be enriched adding information and perspectives on PHCs.

Specific aspects to be covered include:

-       Was there any evaluation of adherence?

-       Was there potential to examine potential drug – drug interactions, ADR or inappropriate use?

-       Could the difference of polypharmacy between the sexes be explained by differential attendance at PHC?  Or to different socio-economic condition? Have you got information regarding frequency of attendance to PHC as a possible marker of poorer control of comorbid conditions/greater polypharmacy?

-       Is there an urban rural discrepancy?

-       Is any information available regarding geriatric syndromes (malnutrition, impaired mobility, falls), nursing home placement, and hospitalizations? It would be of interest to assess the association with polypharmacy, controlled for comorbid conditions

-       Please add a brief description of PHC: are patients attached to an individual physician or is this a “drop in” type system with multiple prescribers? Is the PHC on an electronic medical record viewable by all and are there prescribing reminders according to condition? The authors might usefully refer to Hughes et al., Age Aging, 2013 which examines the potential for polypharmacy in the presence of more than one chronic condition when physicians are adhering to prescribing guidelines

We would appreciate receiving your revised manuscript by 16-JUN-2020. To enhance the reproducibility of your results, we recommend that if applicable you deposit your laboratory protocols in protocols.io, where a protocol can be assigned its own identifier (DOI) such that it can be cited independently in the future. For instructions see: http://journals.plos.org/plosone/s/submission-guidelines#loc-laboratory-protocols

We look forward to receiving your revised manuscript.

Kind regards,

Enrico Mossello and Adrian Stuart Wagg

Academic Editors

PLOS ONE

Journal Requirements:

1.) When submitting your revision, we need you to address these additional requirements:

Please ensure that your manuscript meets PLOS ONE's style requirements, including those for file naming. The PLOS ONE style templates can be found at http://www.plosone.org/attachments/PLOSOne_formatting_sample_main_body.pdf and http://www.plosone.org/attachments/PLOSOne_formatting_sample_title_authors_affiliations.pdf

2.) We note that you have included the phrase “data not shown” in your manuscript. Unfortunately, this does not meet our data sharing requirements. PLOS does not permit references to inaccessible data. We require that authors provide all relevant data within the paper, Supporting Information files, or in an acceptable, public repository. Please add a citation to support this phrase or upload the data that corresponds with these findings to a stable repository (such as Figshare or Dryad) and provide and URLs, DOIs, or accession numbers that may be used to access these data. Or, if the data are not a core part of the research being presented in your study, we ask that you remove the phrase that refers to these data.

3.) In the ethics statement in the manuscript and in the online submission form, please provide additional information about the patient records used in your retrospective study. Specifically, please ensure that you have discussed whether all data were fully anonymized before you accessed them and/or whether the IRB or ethics committee waived the requirement for informed consent. If patients provided informed written consent to have data from their medical records used in research, please include this information.
---

## [Author Response · Author response to Decision Letter 1]

13 May 2020

Response to reviewers

General Response to the Reviewers’ Comments: We are appreciative of the insightful ‎comments raised by the Academic Editors. We believe we have responded to each of their ‎comments. Where necessary, we have made revisions in the manuscript. These revisions are ‎tracked. The line numbers referred to in our responses below refer to those in the tracked ‎version of the manuscript. ‎

ACADEMIC EDITORS:‎

We have concluded that a major revision should be performed for the paper to be ‎published according to Plos One criteria. Moreover a bit more speculation might be useful ‎to suggest strategies for clinical improvement or further investigations.‎

Comment #1: The main point is represented by external validity. It seems clear that the ‎data are not representative of general elderly population in Qatar: prevalence of obesity ‎‎>50%, diabetes almost 80%, asthma 20% are largely greater than population estimates ‎around the world, and may clearly explain the high prevalence of polypharmacy – almost ‎double in comparison with international estimates (to be cited): 39% in US according to ref ‎‎18, 44% in Sweden according to Morin L et al., Clin Epidemiol, 2018). ‎

Response: Thank you for your insightful observation, I agree with you that the prevalence ‎of polypharmacy in our study is higher than some international figures. This could be ‎explained by the high prevalence of NCDs in our study population. We cited Morin on lines ‎‎67 and 206. In the discussion section please see the statement about the association ‎between polypharmacy and prevalence of NCDs (lines 243-244). Our findings are ‎comparable to other regional figures. For example, the prevalence of polypharmacy was ‎‎55% in Saudi Arabia and 60% in Kuwait. Also, a population-based study in Korea found that ‎‎86% of elderly patients were exposed to polypharmacy. ‎

Comment #2: In fact Authors state: "The study population included all Qatari elderly ‎patients (≥65 years) who attended PHC centers during a period of six months from April 1 to ‎September 30, 2017, and who had medication reconciliation done". ‎

What proportion of older Qataris attend PHC? ‎

Response: Thank you for your comment. At the time of the study, the total number of ‎elderly nationals who were attending PHC centers (proven by having active electronic ‎medical records/ encounters) was 15,286. While the total number of the same population in ‎Qatar in 2017 was 17,895 (according to the planning and statistics authority in Qatar). ‎Therefore, this indicates that the proportion of older Qataris who attended PHC centers at ‎the time of the study is almost 85%. Please see additional response in our response to ‎Comment #4.‎

Comment #3: Which percentage of older patients who attended the PHC underwent a ‎medication reconciliation? ‎

Response: According to the Primary Health Care Corporation (PHCC) policy, all patients ‎attending PHC centers must have medication reconciliation. ‎This policy was introduced in ‎January 2017 as a quality measure because patients were seen by any available ‎physicians ‎at the time of their visit‎. Therefore, all patients in our sample have undergone medication ‎reconciliation.‎

Comment #4: Is the sample at least representative of older subjects who attend a Primary ‎Health Center?‎

Response: Thank you for this comment. We believe that our sample is representative of ‎older patients who attend PHC centers because our sample consists of about 37% ‎‎(5,629/15,286) of the total older Qataris who attend PHC centers. In addition, the gender ‎distribution of all elderly Qataris who attended PHC centers across the country in ‎‎2017 was ‎as follows: males= 7526/15286 (49.2%), females= 7760/15286 (50.8%). This is not ‎greatly ‎different from the gender distribution in our sample (i.e., males= 46.2%; females= 53.8%).‎ ‎Therefore, findings from this study provide a reliable basis for the polypharmacy ‎phenomenon in ‎‎primary healthcare settings‎.‎

Comment #5: If this is the case, conclusions should be moderated, stating that the ‎prevalence of polypharmacy of older subjects ATTENDING A PHC in Qatar is very high, and ‎discussion may be enriched adding information and perspectives on PHCs.‎

Response: Thank you for this insightful feedback. We modified the discussion and ‎conclusion sections as advised. Please see line 197 for revisions in the discussion and line ‎‎270-271 in the conclusion. Regarding information and perspectives of PHCs, please see lines ‎‎93 – 100, and 111 – 113.‎

Comment #6: Was there any evaluation of adherence?‎

Response: Thank you for this insightful comment. This is a valid point as our study relied on ‎medications prescribed and reconciled, but not on adherence. To measure adherence ‎would require a dedicated study. We have identified this as a limitation of our study. ‎Please see lines 261 - 265.‎

Comment #7: Was there potential to examine potential drug-drug interactions, ADR or ‎inappropriate use?‎

Response: Thank you for this comment. We did not examine these in our study. However, ‎we (the authors) are currently working on another research to assess the inappropriate use ‎of medications among the same elderly population.‎

Comment #8: Could the difference of polypharmacy between the sexes be explained by ‎differential attendance at PHC? Or to different socio-economic condition? ‎

Response: Thank you for this insightful observation. During the period of our study, the ‎attendance of females was higher than their male counterpart (females= 53.8%; males= ‎‎46.2%). We controlled for this factor in our logistic regression. However, the difference of ‎polypharmacy between the sexes could be explained by the statistically higher prevalence ‎of all NCDs among females than males except for CVD. In addition, the proportion of those ‎who had “three or more NCDs” was higher among females in ‎comparison to males (60.1% ‎vs. 52.3%; p-value <0.001). Also, obesity was more prevalent among females.‎ Please see the ‎new Table 2 in the manuscript. ‎

Comment #9: Have you got information regarding frequency of attendance to PHC as a ‎possible marker of poorer control of comorbid conditions/greater polypharmacy?‎

Response: Thank you for this question. This would have been a valuable information. A ‎research study in Canada showed a significant association between polypharmacy and ‎higher frequency of family physician visits in elderly patients. Please see lines 221-222.‎

Comment #10: Is there an urban rural discrepancy?‎

Response: Thank you for this question. According to PHCC’s Department of Operations, the ‎vast majority of Qatari population 98.75% (15094/15286) who attended PHC centers live in ‎urban areas. This applies to our study population, where around 98.5% lives in urban areas. ‎Thus, we can conclude that there is no urban/rural discrepancy. ‎

Comment #11: Is any information available regarding geriatric syndromes (malnutrition, ‎impaired mobility, falls), nursing home placement, and hospitalizations? It would be of ‎interest to assess the association with polypharmacy, controlled for comorbid conditions.‎

Response: Thank you for this comment. As you will note in our manuscript, we controlled ‎for comorbid conditions in the multivariate logistic regression analysis. Regarding to the ‎other variables you mention, as our data source was the electronic medical records PHC ‎centers, such information is not captured. ‎

Comment #12: Please add a brief description of PHC: are patients attached to an individual ‎physician or is this a “drop in” type system with multiple prescribers? ‎

Response: Thank you for this comment. Please see lines 93 - 96 for a general description of ‎PHCs; and lines 96 – 99 for a specific description of patient/physician assignments. ‎

Comment #13: Is the PHC on an electronic medical record viewable by all and are there ‎prescribing reminders according to condition? ‎

Response: Yes. EMR is viewable by all healthcare professionals who primary care ‎physicians, nurses, and pharmacists. Please see lines 99 – 100. However, there are no ‎prescription renewal reminders according to the health condition.‎

Comment #14: The authors might usefully refer to Hughes et al., Age Aging, 2013 which ‎examines the potential for polypharmacy in the presence of more than one chronic ‎condition when physicians are adhering to prescribing guidelines.‎

Response:‎

Thank you for suggesting the work of Hughes et al. We have referred to his work and ‎another source in lines 265-266.‎

Journal Requirements:‎

‎1.) When submitting your revision, we need you to address these additional requirements: ‎Please ensure that your manuscript meets PLOS ONE's style requirements, including those ‎for file naming. The PLOS ONE style templates can be found at ‎http://www.plosone.org/attachments/PLOSOneformatting_sample_main_body.pdf and

http://www.plosone.org/attachments/PLOSOne_formatting_sample_title_authors_affiliations.pdf

Response: We have checked the style used in our manuscript against those required by ‎PLOS ONE.‎

‎2.) We note that you have included the phrase “data not shown” in your manuscript. ‎Unfortunately, this does not meet our data sharing requirements. PLOS does not permit ‎references to inaccessible data. We require that authors provide all relevant data within the ‎paper, Supporting Information files, or in an acceptable, public repository. Please add a ‎citation to support this phrase or upload the data that corresponds with these findings to a ‎stable repository (such as Figshare or Dryad) and provide and URLs, DOIs, or accession ‎numbers that may be used to access these data. Or, if the data are not a core part of the ‎research being presented in your study, we ask that you remove the phrase that refers to ‎these data.‎

Response: The phrase “Data not shown” is misleading. We meant to say that we are not ‎showing the descriptors in lines 169 – 172 in a table. We have removed this phrase. Please ‎see line 172.‎

‎3.) In the ethics statement in the manuscript and in the online submission form, please ‎provide additional information about the patient records used in your retrospective study. ‎Specifically, please ensure that you have discussed whether all data were fully anonymized ‎before you accessed them and/or whether the IRB or ethics committee waived the ‎requirement for informed consent. If patients provided informed written consent to have ‎data from their medical records used in research, please include this information.‎

Response: Please see lines 139 – 141 where we have added this required information.‎

---

## [Editor Report · Decision Letter 2]

27 May 2020

Prevalence of polypharmacy and the association with non-communicable diseases in Qatari elderly patients attending primary healthcare centers: A cross-sectional study

PONE-D-19-22287R2

Dear Dr. Ayman Al Dahshan,

We are pleased to inform you that your manuscript has been judged scientifically suitable for publication and will be formally accepted for publication once it complies with all outstanding technical requirements.

With kind regards,

Enrico Mossello

Academic Editor

PLOS ONE
---

## [Editor Report · Acceptance letter]

1 Jun 2020

PONE-D-19-22287R2 

Prevalence of polypharmacy and the association with non-communicable diseases in Qatari elderly patients attending primary healthcare centers: A cross-sectional study 

Dear Dr. Al Dahshan:

I am pleased to inform you that your manuscript has been deemed suitable for publication in PLOS ONE. Congratulations! Your manuscript is now with our production department. 

With kind regards,

on behalf of

Dr. Satya Surbhi 

Academic Editor

PLOS ONE